# Evaluation of Long-Read Sequencing Simulators to Assess Real-World Applications for Food Safety

**DOI:** 10.3390/foods13010016

**Published:** 2023-12-19

**Authors:** Katrina L. Counihan, Siddhartha Kanrar, Shannon Tilman, Andrew Gehring

**Affiliations:** Eastern Regional Research Center, United States Department of Agriculture, Agricultural Research Service, Wyndmoor, PA 19038, USA; siddhartha.kanrar@usda.gov (S.K.); shannon.tilman@usda.gov (S.T.); andrew.gehring@usda.gov (A.G.)

**Keywords:** Shiga toxin-producing *Escherichia coli* O157:H7, *Listeria monocytogenes*, *Bos taurus*, foodborne pathogens, virulence genes

## Abstract

Shiga toxin-producing *Escherichia coli* (STEC) and *Listeria monocytogenes* are routinely responsible for severe foodborne illnesses in the United States. Current identification methods utilized by the U.S. Food Safety Inspection Service require at least four days to identify STEC and six days for *L. monocytogenes*. Adoption of long-read, whole genome sequencing for food safety testing could significantly reduce the time needed for identification, but method development costs are high. Therefore, the goal of this project was to use NanoSim-H software to simulate Oxford Nanopore sequencing reads to assess the feasibility of sequencing-based foodborne pathogen detection and guide experimental design. Sequencing reads were simulated for STEC, *L. monocytogenes*, and a 1:1 combination of STEC and *Bos taurus* genomes using NanoSim-H. At least 2500 simulated reads were needed to identify the seven genes of interest targeted in STEC, and at least 500 reads were needed to detect the gene targeted in *L. monocytogenes*. Genome coverage of 30x was estimated at 21,521, and 11,802 reads for STEC and *L. monocytogenes*, respectively. Approximately 5–6% of reads simulated from both bacteria did not align with their respective reference genomes due to the introduction of errors. For the STEC and *B. taurus* 1:1 genome mixture, all genes of interest were detected with 1,000,000 reads, but less than 1x coverage was obtained. The results suggested sample enrichment would be necessary to detect foodborne pathogens with long-read sequencing, but this would still decrease the time needed from current methods. Additionally, simulation data will be useful for reducing the time and expense associated with laboratory experimentation.

## 1. Introduction

Foodborne pathogens remain a major health concern and economic burden in the United States. The Centers for Disease Control estimates that 48 million people are sickened by foodborne illnesses each year, with 128,000 hospitalizations [1]. Many of those hospitalizations are due to infection by Shiga toxin-producing *Escherichia coli* (STEC) and *Listeria monocytogenes* [1]. In the U.S. in 2021, pathogen contamination resulted in 47 meat recalls totaling over 15 million pounds of meat; two recalls were due to STEC contamination and five due to *Listeria* [2]. It is estimated that foodborne disease costs the U.S. economy approximately $17 billion annually [3].

Infection with STEC can result in hemorrhagic colitis and hemolytic uremic syndrome [4]. Serotypes of *E. coli* are determined based on the polysaccharide O-antigen in the lipopolysaccharide outer membrane and the H-antigen on the flagella [5]. The STEC serotype most frequently associated with outbreaks is O157:H7 [1]. The U.S. Department of Agriculture Food Safety and Inspection Service (USDA FSIS) isolates and identifies STEC in meat through a combination of culturing, molecular methods, O typing, and matrix-assisted laser desorption/ionization time-of-flight (MALDI-TOF) mass spectrometry [6], and the process takes four days to complete. Confirmation of *E. coli* O157:H7 is through the detection of the virulence genes *eae*, *stx1*, *stx2*, and *fliC*, and the ribosomal 16S rRNA gene, *rrsC*, using quantitative PCR (qPCR) [6]. The *eae* gene product is intimin, which mediates enterocyte colonization [4], and *fliC* encodes the flagellar H-antigen determinant [7]. Expression of genes *stx1* and *2* produces Shiga toxin 1 and 2, respectively, which are responsible for surface localization and cytotoxicity [4]. The U.S. Food and Drug Administration’s (FDA) method to test other food and beverages for *E. coli* takes at least 5 days and involves an RT-PCR screen, culture confirmation, antisera testing, RT-PCR confirmation testing, and characterization with pulsed-field gel electrophoresis and whole genome sequencing [8].

Listeriosis, caused by *L. monocytogenes*, generally results in gastroenteritis, but some cases result in sepsis, meningitis, or, in pregnant women, fetal infection [9]. Isolation and confirmation of *L. monocytogenes* from meat, egg, or environmental samples by FSIS involves a combination of culturing, molecular methods, and MALDI-TOF mass spectrometry and takes six days to complete [10]. Molecular identification methods generally target the *hly* gene, which encodes the hemolysin, listeriolysin O [11]. The FDA method involves culture isolation, biochemical or RT-PCR confirmation, and serological and genetic subtyping and takes at least 5 days, but likely more depending on the tests selected and incubation times needed [12].

Advances in whole genome sequencing technology have led to third-generation, or long-read, sequencing that could significantly reduce the amount of time needed to identify foodborne pathogens from current culture-based methods. Oxford Nanopore Technologies’ MinION device sequences RNA or DNA by detecting changes in electrical current as the strands of nucleic acid pass through nanopores on a flow cell [13]. Long reads are generated that facilitate genome assembly [14], and real-time analysis allows pathogen detection to be accomplished in hours instead of days if samples are sequenced directly [15]. Including a 24-h growth enrichment prior to sequencing may enhance detection, and this would still reduce the time needed to positively identify samples from current methods. Additionally, the small, portable sequencers allow whole genome analysis to be conducted outside of traditional laboratories, and the cost is generally lower than second-generation sequencing. Whole genome sequencing is also advantageous for serotype resolution and antibiotic resistance monitoring. However, a disadvantage of nanopore sequencing is that it is more error-prone, although accuracy is rapidly improving [15].

Simulator software has been developed to assist with the planning and data analysis of long-read sequencing experiments. NanoSim was developed to simulate Oxford Nanopore reads and error rates [16], and a derivation of the program, NanoSim-H, incorporates several improvements and bug fixes [17]. The software operates in two steps. First, the input genome is characterized by an alignment-based analysis, and a model is produced that incorporates errors and length distributions. Second, the model is used to simulate sequencing run reads [16]. While nanopore sequencing is generally more affordable and faster than second-generation sequencing, novel method development can be expensive and time-consuming. Simulation results can guide method development to reduce the time and cost of benchwork. Therefore, the objectives of this project were to determine the number of reads needed to detect target genes in STEC and *L. monocytogenes* and evaluate the influence of host genetic material on detection. The goal of this project was to use the simulation data to assess the feasibility of using long-read sequencing for metagenomic analysis of samples to detect foodborne pathogens without enrichment and to guide experimental design.

## 2. Materials and Methods

### 2.1. Sequencing Read Simulation

Oxford Nanopore sequencing reads were simulated using reference genomes for pathogenic human strains of *E. coli* O157:H7 (NCBI Accession #NC002695.2) and *L. monocytogenes* (NCBI Accession #NC003210.1) and a combination of an *E. coli* O157:H7 genome and a *Bos taurus*, domestic Hereford bovine, genome (Accession #NC037338.1) using NanoSim-H (Version 1.1.0.4). Details on the development and functioning of the NanoSim-H program are in Yang et al. [16] and Brinda et al. [17]. Briefly, NanoSim-H was run with Python (version 3.6) using default parameters, except for the -n (number of reads) argument, which was changed to specify how many reads were output. Whole genome assemblies downloaded from NCBI (accession numbers above) were input as fasta files. The STEC genome was input for STEC simulations, the *L. monocytogenes* genome was input for *L. monocytogenes* simulations, and one STEC and one *B. taurus* genome were combined into one fasta file for input into the program for the STEC and bovine mix simulations. For the STEC and *L. monocytogenes* simulations, the circular argument was also used to specify the simulation of circular genomes. The following number of reads (-n argument) were simulated for STEC, *L. monocytogenes*, and the STEC/bovine combination: 10, 50, 75, 100, 250, 500, 750, 1000, 2500, 5000, 7500, 10,000, 50,000, 75,000, 100,000, 250,000, 500,000, 750,000, 1,000,000. The fasta files of the simulated reads are available on the Ag Data Commons (https://doi.org/10.15482/USDA.ADC/1529447, deposited on 29 August 2023). The fasta file generated from each simulated read set was analyzed using SeqKit software (version 2.2.0, [18]) to obtain the following information: read N50, read N90, minimum read length (b), and maximum read length (b). Only one simulation per -n argument was analyzed because the computer program produced the same output when the same -n argument was simulated.

### 2.2. De-Novo Assembly

The reads generated were used for de novo genome assembly using Flye software (version 2.8.3-b1695, [19]) with default parameters for nanopore raw reads and genome sizes of 5.5 Mb for STEC and 3 Mb for *L. monocytogenes*. The Flye assembly output file was further analyzed with SeqKit software to determine total assembly length (b), number of contigs, smallest contig length (b), maximum contig length (b), genome coverage, and assembly N50 and N90.

### 2.3. Virulence Gene Identification

The following genes were targeted for detection in the simulations with STEC alone and in combination with the bovine genome: *fliC*, *eae*, *stx1a*, *stx1b*, *stx2a*, *stx2b*, *rrsC*. The *hly* gene was targeted in *L. monocytogenes*. Reads less than 200 bp were removed from the analysis. The reads were aligned against the respective genome (STEC or *L. monocytogenes*) in Geneious Prime software (version 2022) using Minimap2. The target genes were searched for in the Annotations and Tracks tab. In the Graphs tab, the Highlight Above function was used to determine how many copies of the target genes were aligned to the reference.

## 3. Results

### 3.1. Escherichia coli O157:H7

NanoSim-H was used to generate simulated reads from the STEC genome, and the simulated read characteristics and genome assembly data are summarized in Table 1. The N50 and N90 were similar for most sets of simulated reads, approximately 9400 and 5200, respectively. Read lengths ranged from 45–66,949 b. De novo genome assembly was possible starting with 1000 reads and generated three contigs with an assembly N50 of 117,920 b and 1x genome coverage. The 1,000,000 simulated reads were also assembled into three contigs, but the assembly N50 was 5,054,755 b, and 1411x genome coverage was obtained; 30x coverage occurred between 10,000 and 50,000 reads. The number of times each gene of interest, fliC, stx1a, stx1b, stx2a, stx2b, eae, and rrsC, was detected for each set of reads is reported in Table 2. Some virulence gene detection started with 50 simulated reads, but all seven genes of interest were not detected together until 2500 reads were simulated. Approximately 5–6% of reads could not be aligned to the reference genome for all sets of simulated reads except for 10 simulated reads, where 10% of the reads could not be aligned.

### 3.2. Listeria Monocytogenes

Simulated reads were generated for *L. monocytogenes* using NanoSim-H, and Table 3 details the simulated read characteristics and genome assembly data. The N50 and N90 were slightly lower than STEC, averaging 9300 and 5200, respectively. The shortest read length was 38 b, and the longest was 66,752 b. De novo genome assembly occurred starting with 750 reads and generated four contigs with an assembly N50 of 28,222 b and 1x genome coverage. The 1,000,000 simulated reads were assembled into two contigs with an N50 of 2,934,114 b and 2583x genome coverage; 30x coverage was obtained around 10,000 reads. Table 4 details the number of times the gene of interest, *hly*, was detected for each set of reads. Detection of *hly* first occurred with 500 simulated reads, and there was over 10x detection of *hly* with 5000 simulated reads. Approximately 5–6% of reads could not be aligned to the reference genome for all simulated read sets, except for 10 and 50, which had 10% and 8% of their reads, respectively, not aligned.

### 3.3. Escherichia coli O157:H7 and Bos taurus

NanoSim-H was used to simulate reads from a 1:1 ratio of STEC:*Bos taurus* genomes and the read characteristics are provided in Table 5. The N50 and N90 were approximately 9300 and 5100, respectively, for most reads. The read lengths were between 34 b and 66,883 b. De-novo assembly of the STEC genome was only achieved with 750,000 and 1,000,000 simulated reads, and coverage was less than 1x. A 25,450 b contig was assembled with 750,000 reads. With 1,000,000 reads, 36 contigs were assembled with a total length of 1,318,433 b, an N50 of 34,338, and 0.93 genome coverage. The smallest contig was 1333 b, and the largest was 98,223 b. Table 6 shows how many times each STEC gene of interest was detected for each set of simulated reads. Genes were first detected with 7500 simulated reads, but not all seven genes were detected simultaneously until 1,000,000 reads were simulated. All reads for the 10, 50, 75, 250, and 750 read simulations did not align with the STEC genome, and approximately 99% of reads for the remaining simulations did not align.

## 4. Discussion

The results of this project provide guidance for the development of foodborne pathogen detection programs using MinION sequencing. NanoSim-H software, an updated version of the NanoSim simulator, was used to simulate ONT sequence data because of its convenience and specificity. NanoSim is a rapid, scalable simulator of ONT sequencing technology-specific data that can be modified as ONT technology improves. NanoSim-H can run in R or Python. Additionally, NanoSim-H has been shown to simulate error events, fragment lengths, and alignment ratios of ONT reads more accurately than other simulation programs [16]. NanoSim was benchmarked with DNA prepared using sequencing kits that fragment the DNA [16], and simulation results are representative of using an ONT Rapid or Field Sequencing Kit. The program can be trained to provide simulated results with different library preparation kits, but as the Rapid and Field Sequencing Kits are simple and require minimal equipment, they would be the first choice for use in food safety testing. Therefore, NanoSim-H was used as benchmarked. The number of reads needed to detect virulence genes and to provide sufficient genome coverage was determined for STEC and *L. monocytogenes*. For STEC, all seven virulence genes of interest were identified beginning with 2500 simulated reads, while the *L. monocytogenes* virulence gene of interest, *hly*, was detected starting with 500 simulated reads.

Genome coverage of 30x is desired for high-quality assemblies to ensure all regions of the genome are sequenced at least once, and that sequence variations can be distinguished from errors [20]. However, for pathogen identification, 10x coverage would be sufficient as error rates were 5–6% (discussed more fully below). For STEC, coverage of 14x was obtained with 10,000 reads, and each virulence gene of interest was detected an average of 12 times, which would allow confident identification (Figure 1). With 50,000 reads, 70x coverage was observed, and each virulence gene was identified an average of 61 times. A linear regression analysis indicated 30x coverage for a high-quality assembly would be obtained with 21,521 reads. For *L. monocytogenes*, coverage of 12x was obtained with 5000 reads, and the virulence gene of interest was detected 15 times (Figure 1). With 50,000 reads, coverage increased to 129x, and *hly* was detected 124 times. A linear regression suggested 11,802 reads would provide 30x coverage. Fewer reads were needed to obtain more coverage of the *L. monocytogenes* genome (2.9 Mb) because it is nearly half the size of the STEC genome (5.5 Mb). Additionally, only one gene was targeted in *L. monocytogenes*, while seven genes were targeted in STEC (Table 7). These results suggest that the number of sequences, and therefore time, needed to detect pathogens will depend on the genome size and number of genes targeted. Bacteria with smaller genomes or with fewer virulence genes of interest should require shorter sequencing times.

A simulation of MinION sequencing reads using a 1:1 ratio of STEC to bovine genomes was conducted, and predictably, adding the bovine genome made detecting STEC virulence genes of interest more difficult. Even with 1,000,000 simulated reads, 1x coverage of the STEC genome was not achieved, and the genes of interest were only detected an average of three times each. The majority of reads simulated with the mixed genomes did not align with the STEC reference. This is not surprising, considering the bovine genome is approximately 490 times larger than the STEC genome and would have a higher probability of being sequenced. In actual meat samples, the amount of bovine DNA in comparison to pathogen DNA would far exceed the 1:1 ratio used in this study. However, running simulations with a higher ratio of bovine to STEC DNA was not undertaken in this study due to the huge computing power and time required for such an experiment. Therefore, the results were scaled up mathematically to provide an estimate of the amount of sequencing time needed to detect STEC in a meat sample. The current FSIS protocol tests 325 g samples of raw ground beef for STEC [6]. An estimated 1.08 × 10^11^ bovine cells would be in 325 g of beef, assuming a mammalian cell mass of 3 ng [21,22]. The STEC genome is approximately 5.5 Mb, and the bovine genome is 2711 Mb; therefore, one STEC genome in the 325 g of meat would represent only 9.23 × 10^−10^% of the genomes in the sample. The average size of simulated reads was 9444 bp, and based on other studies [23,24,25], an average of 80,638 reads per hour could be expected. This suggests that 7.82 × 10^8^ h of sequencing would be needed to obtain 1x coverage of the STEC genome. Therefore, obtaining 10x coverage to ensure there are no false negatives in detection would substantially increase the amount of time needed to 2.35 × 10^10^ h. This suggests that detection of foodborne pathogens with MinION sequencing would be impractical without enrichment for the bacteria of interest.

Food samples are often enriched to multiply pathogen numbers and increase the probability of detection. In the current FSIS protocols, STEC samples are enriched in modified tryptone soy broth for 15–24 h [6], which adds one day to the testing regime. *L. monocytogenes* undergoes primary enrichment in modified University of Vermont broth for 20–26 h and then secondary enrichment in (3-N-morpholino) propanesulfonic acid-buffered *Listeria* enrichment broth for 18–24 h [10], adding two days to the protocol. Developing a method to detect pathogens without enrichment is a major goal, but even with enrichment, MinION sequencing could still reduce the amount of time needed to identify pathogens in a sample. Enrichment and plating would take two days, and DNA extraction, sequencing, and data analysis could be conducted on the third day. This would result in species confirmation one day faster than the current FSIS STEC protocol and three days faster than the *L. monocytogenes* protocol. This time savings would allow meat products to arrive at the market more quickly, reducing losses due to spoilage.

A disadvantage of third-generation sequencing compared to second-generation is the higher error rate in sequencing data. Error rates of 5–20% have been reported [26,27]. In this study, the error rate was on the lower side of the reported estimates, with 5–6% of reads simulated from STEC or *L. monocytogenes* genomes not aligning with their respective reference genomes due to the introduction of errors. Errors (mismatches, insertions, and deletions) were determined by the NanoSim-H program using statistical mixture models [16]. Despite these inaccuracies, the simulated sequencing data were sufficient to identify the genes of interest in STEC and *L. monocytogenes*. Additionally, the error rate of MinION sequencing may be substantially improving as the latest MinION flow cells (version R10.4.1) are supposed to deliver accuracy above 99% [13,28], which is on par with next-generation sequencing platforms that have error rates of less than 1% [29]. ONT also released the Dorado basecaller in 2023, which is the newest version of the program that converts electrical disruptions into basecalls. The increased speed of Dorado will allow the use of higher accuracy basecalling models in real-time and further reduce error rates [30]. Improved accuracy in long-read sequencing would be beneficial for differentiating between serotypes in bacteria, such as *Salmonella*, which vary by a single nucleotide polymorphism [31] and where inaccurate base calling would be problematic. Conventional methods of serotyping with antisera are time intensive, and sequencing could significantly reduce the time needed for identification.

Short-read sequencing with platforms, such as Illumina, have also been evaluated for foodborne pathogen detection, particularly in produce [32,33,34]. Pathogens could be successfully identified, but short-read sequencing has disadvantages compared to long-read sequencing in rapid testing methods. It requires longer, more labor-intensive library preparation and larger equipment, which would make on-site applications unlikely and may be cost-prohibitive [35]. Additionally, real-time analysis cannot be conducted with short-read sequencing, increasing the amount of time needed for pathogen identification [35]. The MinION platform will also sequence any DNA in the library, both short and long DNA fragments, maximizing the sequencing data obtained. A challenge for all sequencing platforms will be the low quantity of pathogen DNA as compared to the host DNA. Methods will need to be optimized to address this issue during sample preparation prior to sequencing and during data analysis.

There were a few limitations in this study that could be addressed in future research. First, we were unable to obtain replicate sequences for a sample because the simulator would output the same set of reads for a particular genome input. Real sequencing runs would vary, even with replicate runs on the same DNA extraction, because only a portion of the sample is sequenced, and the genomic fragments available for sequencing would be different between runs, especially in metagenomic samples. It is possible that coding changes could allow for varied outputs. Another limitation was the size of the fasta files that could be processed. As noted above, we only simulated a 1:1 *E. coli*:bovine mix because of the intense computing power required, even using the U.S. Department of Agriculture high-performance computing system. As computational power advances, high-performance computers may be able to run these simulations, which would allow more realistic genome mixtures to be analyzed. We also limited this study to one *E. coli* and one *Listeria* serotype. However, food samples may be contaminated with multiple serotypes of one bacterial species or multiple bacterial species. Future simulation studies are needed to assess how species and serotypes can be differentiated.

## 5. Conclusions

The results of this project will be used to guide laboratory experiments to develop foodborne pathogen screening methods using MinION sequencing, reducing the time, materials, and expense needed. The in silico generation of sequencing reads with NanoSim-H suggested that enrichment will be necessary for MinION sequencing to be practical because the overwhelming amount of host DNA in the sample reduces the detection of the smaller concentrations of pathogen DNA. However, even with enrichment, MinION sequencing could decrease the amount of time needed to identify *E. coli* by one day and *L. monocytogenes* by three days as compared to current FSIS methods. Additionally, samples from multiple enrichments selecting for different pathogens could be barcoded and run on one MinION flow cell, which would save time and be more cost-effective. MinION sequencing is also less labor intensive than culture-based methods, which would decrease staff time. This time savings would result in products reaching the market faster and reduce loss due to spoilage. MinION sequencing shows promise as a food pathogen testing method, and simulation data will be a useful tool for efficient experimental design.

## Figures and Tables

**Figure 1 foods-13-00016-f001:**
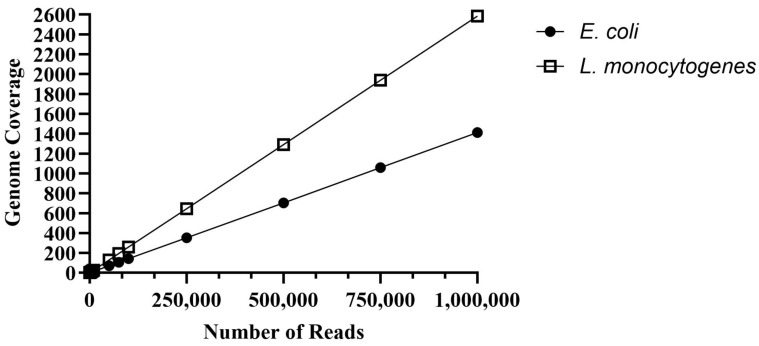
The amount of genome coverage versus the number of simulated reads for *Escherichia coli* and *Listeria monocytogenes*.

**Table 1 foods-13-00016-t001:** Statistics and de novo genome assembly data for each set of simulated reads for *Escherichia coli* O157:H7.

Number of Reads	Reads N50 (b)	Reads N90 (b)	Minimum Read Length (b)	Maximum Read Length (b)	Total Read Length (b)	Assembly	Contig Number in Assembly	Total Length (b)	Genome Coverage	Assembly N50 (b)	Smallest Contig (b)	Largest Contig (b)
10	8849	3483	2582	13,424	72,759	No						
50	10,049	5788	208	33,863	422,715	No						
75	9233	5346	208	15,536	540,757	No						
100	9375	5040	221	17,938	757,508	No						
250	9615	5317	217	38,181	1,971,775	No						
500	9624	5078	125	43,121	3,878,109	No						
750	9700	5476	141	28,563	5,987,151	No			1			
1000	9577	5462	141	30,045	7,932,184	Yes	3	185,255	1	117,920	18,111	117,920
2500	9531	5319	94	40,056	19,646,989	Yes	42	2,972,053	3	98,451	26,055	163,256
5000	9426	5232	100	43,248	38,804,704	Yes	27	5,547,668	6	377,010	16,126	614,371
7500	9348	5366	72	45,099	58,630,216	Yes	14	5,525,979	10	1,120,145	12,681	1,327,415
10,000	9407	5282	87	58,762	78,143,078	Yes	9	5,472,930	14	4,015,755	17,274	4,015,755
50,000	9412	5241	81	59,173	388,401,901	Yes	9	5,519,291	70	3,781,995	17,242	3,781,995
75,000	9369	5280	57	59,494	582,224,483	Yes	7	5,489,935	106	4,015,794	17,597	4,015,794
100,000	9380	5280	53	59,146	778,028,874	Yes	7	5,537,798	141	4,013,828	17,324	4,013,828
250,000	9384	5263	49	58,657	1,939,653,454	Yes	6	5,485,282	352	4,029,574	164,347	4,029,574
500,000	9380	5256	50	66,523	3,874,545,789	Yes	4	5,505,839	704	4,013,857	278,922	4,013,857
750,000	9384	5258	48	66,949	5,819,660,420	Yes	5	5,469,961	1058	4397,660	163,836	4,397,660
1,000,000	9389	5271	45	66,154	7,764,565,277	Yes	3	5,479,648	1411	5,054,755	166,955	5,054,755

**Table 2 foods-13-00016-t002:** Number of times the genes of interest were detected for each set of *Escherichia coli* simulated reads.

Number of Reads	fliC	stx1A	stx1B	stx2A	stx2B	eae	rrsC
10	0	0	0	0	0	0	0
50	1	0	0	0	0	0	0
75	0	1	1	0	0	0	0
100	0	1	1	0	0	0	0
250	0	0	0	1	1	1	1
500	0	0	0	0	0	1	1
750	2	0	1	3	3	1	2
1000	3	0	0	1	2	2	1
2500	5	3	3	4	4	5	3
5000	10	9	6	2	2	9	7
7500	11	13	13	14	14	3	15
10,000	15	14	11	8	7	14	13
50,000	69	48	45	68	61	56	54
75,000	109	99	96	92	91	108	96
100,000	126	132	132	128	133	122	130
250,000	291	315	289	320	314	334	287
500,000	717	593	576	651	644	650	661
750,000	939	929	901	962	917	970	896
1,000,000	1293	1310	1286	1269	1236	1260	1304

**Table 3 foods-13-00016-t003:** Statistics and de novo genome assembly data for each set of simulated reads for *Listeria monocytogenes*.

Number of Reads	Reads N50 (b)	Reads N90 (b)	Minimum Read Length (b)	Maximum Read Length (b)	Total Read Length (b)	Assembly	Contig Number in Assembly	Total Length (b)	Genome Coverage	Assembly N50 (b)	Smallest Contig (b)	Largest Contig (b)
10	8494	4740	1427	12,559	72,583	No						
50	9577	5642	210	41,463	430,631	No						
75	9439	5291	223	19,182	584,397	No						
100	9106	5388	234	23,065	794,250	No						
250	10,076	5075	193	30,465	1,927,189	No						
500	9486	5178	124	36,891	3,872,406	No			1			
750	9494	5271	170	42,201	5,818,869	Yes	4	114,800	1	28,222	25,967	32,938
1000	9131	4938	146	29,164	7,529,374	Yes	10	256,060	2	29,255	6012	41,172
2500	9403	5344	104	32,780	19,425,020	Yes	55	2,949,097	6	67,244	5924	150,384
5000	9336	5265	106	44,643	38,786,230	Yes	7	2,953,689	12	1,646,363	9293	1,646,363
7500	9343	5302	70	46,897	58,317,980	Yes	2	2,941,344	19	2,936,243	5101	2,936,243
10,000	9336	5268	80	58,755	77,468,040	Yes	1	2,932,595	25			
50,000	9371	5261	71	58,893	388,057,432	Yes	2	2,938,525	129	2,932,511	6,014	2,932,511
75,000	9369	5263	65	58,824	581,362,326	Yes	2	2,938,503	193	2,932,489	6014	2,932,489
100,000	9389	5267	56	59,335	776,241,829	Yes	2	2,938,504	258	2,932,485	6019	2,932,485
250,000	9359	5253	47	58,406	1,936,081,693	Yes	2	2,938,523	645	2,932,499	6024	2,932,499
500,000	9361	5252	48	66,752	3,873,185,429	Yes	2	2,940,270	1291	2,934,255	6015	2,934,255
750,000	9373	5259	38	66,124	5,817,330,686	Yes	2	2,927,521	1939	2,912,918	14,603	2,912,918
1,000,000	9370	5260	49	66,194	7,751,957,440	Yes	2	2,940,128	2583	2,934,114	6014	2,934,114

**Table 4 foods-13-00016-t004:** Number of times the gene of interest was detected for each set of *Listeria monocytogenes* simulated reads.

Number of Reads	hly
10	0
50	0
75	0
100	0
250	0
500	2
750	1
1000	4
2500	8
5000	15
7500	22
10,000	18
50,000	124
75,000	164
100,000	266
250,000	604
500,000	1182
750,000	1737
1,000,000	2463

**Table 5 foods-13-00016-t005:** Statistics for each set of simulated reads in the 1:1 *Escherichia coli*:bovine genome mix.

Number of Reads	N50	N90	Minimum Length (b)	Maximum Length (b)	Total Read Length (b)
10	10,843	6084	3548	16,886	96,456
50	7887	4469	203	18,360	335,562
75	9463	4519	213	25,060	537,965
100	9451	4293	228	18,234	714,103
250	9083	5442	209	41,628	1,938,635
500	9208	5092	140	25,845	3,790,451
750	9175	5132	146	32,043	5,549,829
1000	9512	5351	124	24,760	7,921,039
2500	9625	5363	110	42,995	19,755,086
5000	9286	5210	107	40,488	38,488,335
7500	9341	5251	65	43,911	58,047,341
10,000	9370	5323	54	58,712	77,620,898
50,000	9367	5221	90	58,855	386,710,714
75,000	9391	5244	58	58,685	582,821,152
100,000	9370	5264	71	58,685	776,242,471
250,000	9368	5267	44	58,866	1,937,593,079
500,000	9376	5260	43	66,488	3,877,065,482
750,000	9362	5262	48	66,883	5,809,149,197
1,000,000	9368	5250	34	66,412	7,747,092,280

**Table 6 foods-13-00016-t006:** Number of times each gene of interest was detected for the simulated reads in the 1:1 *Escherichia coli*:bovine genome mix.

Number of Reads	fliC	stx1A	stx1B	stx2A	stx2B	eae	rrsC
10	0	0	0	0	0	0	0
50	0	0	0	0	0	0	0
75	0	0	0	0	0	0	0
100	0	0	0	0	0	0	0
250	0	0	0	0	0	0	0
500	0	0	0	0	0	0	0
750	0	0	0	0	0	0	0
1000	0	0	0	0	0	0	0
2500	0	0	0	0	0	0	0
5000	0	0	0	0	0	0	0
7500	0	1	1	0	0	0	0
10,000	0	0	0	0	0	0	0
50,000	0	0	0	0	0	0	0
75,000	1	0	0	1	1	0	1
100,000	0	0	0	2	2	2	0
250,000	1	0	0	1	1	1	0
500,000	2	0	0	3	3	1	2
750,000	3	4	4	3	3	0	5
1,000,000	3	1	2	5	5	3	2

**Table 7 foods-13-00016-t007:** Genome and gene of interest lengths for *Escherichia coli* O157:H7 and *Listeria monocytogenes*.

Bacteria	Genome Length (b)	Gene Name	Length (b)
*E. coli*	5,498,578	fliC	1758
		eae	2805
		rrsC	1538
		stx1A	948
		stx1B	270
		stx2A	960
		stx2B	270
*L. monocytogenes*	2,944,528	hly	1590

## Data Availability

The datasets generated for this study can be found in the Ag Data Commons: https://doi.org/10.15482/USDA.ADC/1529447, deposited 29 August 2023.

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
