# Peer review of "Evaluation of Long-Read Sequencing Simulators to Assess Real-World Applications for Food Safety"

_foods, 2023, doi:10.3390/foods13010016_

Round 1

Reviewer 1 Report

Comments and Suggestions for Authors

The manuscript entitled "Use of Long-Read Sequencing Simulators to Assess Real-World Applications for Food Safety" presents a strategy for utilizing NanoSim-H software to simulate Oxford Nanopore sequencing reads to assess the feasibility of sequencing-based foodborne pathogen detection and guide experimental design. After careful evaluation, I have identified several areas that require revisions before the manuscript can be considered for publication.

1.     Address potential limitations and challenges: Acknowledge any limitations or challenges encountered during the study.

2.     Provide suggestions for potential solutions or future research directions to overcome these limitations.

Comments on the Quality of English Language

Proofread and improve clarity: Review the manuscript for grammar, spelling, and sentence structure errors.

Author Response

Responses to the Comments from Reviewer #1

Summary

Thank you very much for your comments. We appreciate the time taken to review the manuscript. They have been thoughtfully addressed and will make the manuscript stronger. Our response to each is detailed below.

Questions for General Evaluation

Quality of English Language

( ) I am not qualified to assess the quality of English in this paper
( ) English very difficult to understand/incomprehensible
( ) Extensive editing of English language required
( ) Moderate editing of English language required
(x) Minor editing of English language required
( ) English language fine. No issues detected

Response and Revisions: Please see the section below, “Comments on the Quality of English Language”, for our response.

Yes

Can be improved

Does the introduction provide sufficient background and include all relevant references?

(x)

( )

Are all the cited references relevant to the research?

(x)

( )

Is the research design appropriate?

( )

(x)

Are the methods adequately described?

( )

(x)

Are the results clearly presented?

(x)

( )

Are the conclusions supported by the results?

(x)

( )

Response and Revisions

Limitations of the research design and possible solutions to these challenges in future studies were addressed. We also added more details in the methods section.

Point-by-Point Response to Comments and Suggestions

The manuscript entitled "Use of Long-Read Sequencing Simulators to Assess Real-World Applications for Food Safety" presents a strategy for utilizing NanoSim-H software to simulate Oxford Nanopore sequencing reads to assess the feasibility of sequencing-based foodborne pathogen detection and guide experimental design. After careful evaluation, I have identified several areas that require revisions before the manuscript can be considered for publication.

Comment 1: Address potential limitations and challenges: Acknowledge any limitations or challenges encountered during the study.

Response 1: A paragraph was added that addresses limitations, challenges, and potential solutions to these limitations in future studies.

Pg. 14 – 15, Lines 294 - 308 “There were a few limitations in this study that could be addressed in future research. First, we were unable to obtain replicate sequences for a sample because the simulator would output the same set of reads for a particular genome input. Real sequencing runs would vary, even with replicate runs on the same DNA extraction, because only a portion of the sample is sequenced and the genomic fragments available for sequencing would be different between runs, especially in metagenomic samples. It is possible that coding changes could allow for varied outputs. Another limitation was the size of the fasta files that could be processed. As noted above, we only simulated a 1:1 E. coli:bovine mix because of the intense computing power required, even using the U.S. Department of Agriculture high-performance computing system. As computational power advances, the high-performance computers may be able to run these simulations, which would allow more realistic genome mixtures to be analyzed. We also limited this study to one E. coli and one Listeria serotype. However, food samples may be contaminated with multiple serotypes of one bacterial species or multiple bacterial species. Future simulation studies will be needed to assess how species and serotypes can be differentiated.”

Comment 2: Provide suggestions for potential solutions or future research directions to overcome these limitations.

Response 2: Please see Response 1 above.

Comments on the Quality of English Language

Comment 1: Proofread and improve clarity: Review the manuscript for grammar, spelling, and sentence structure errors.

Response 1: The manuscript was carefully reviewed to correct grammar, spelling, and sentence structure errors and to improve clarity.

Reviewer 2 Report

Comments and Suggestions for Authors

Dear Authors,

The manuscript Use of Long-Read Sequencing Simulators to Assess Real-World Applications for Food Safety was investigated. Please, find below my comments:

-The topic of this research is of great interest, since the use NanoSim-H software to to sequence and reduce identification time of each pathogen

-In the material and method section : You should utilize subheadings to divide up different subsections. 

-You should detail the procedure of enrichissement of pathogens numbers

-In the figure 1, avoid making symbols in title Please change ‘ #of reads’ by ‘number of reads’

-Italicize bacterial strain in all the manuscript

Author Response

Responses to the Comments from Reviewer #2

Summary

Thank you very much for your comments. We appreciate the time taken to review the manuscript. They have been thoughtfully addressed and will make the manuscript stronger. Our response to each is detailed below.

Questions for General Evaluation

Quality of English Language

(x) I am not qualified to assess the quality of English in this paper
( ) English very difficult to understand/incomprehensible
( ) Extensive editing of English language required
( ) Moderate editing of English language required
( ) Minor editing of English language required
( ) English language fine. No issues detected

Yes

Can be improved

Must be improved

Does the introduction provide sufficient background and include all relevant references?

(x)

( )

( )

Are all the cited references relevant to the research?

( )

(x)

( )

Is the research design appropriate?

( )

(x)

( )

Are the methods adequately described?

( )

( )

(x)

Are the results clearly presented?

(x)

( )

( )

Are the conclusions supported by the results?

(x)

( )

( )

Response and Revisions

We have added a discussion on the limitations of the research design and ways to address these challenges in future studies. Additional details were added to the methods to further clarify what was done.

Point-by-Point Response to Comments and Suggestions

Dear Authors,

The manuscript Use of Long-Read Sequencing Simulators to Assess Real-World Applications for Food Safety was investigated. Please, find below my comments:

Comment 1: The topic of this research is of great interest, since the use NanoSim-H software to to sequence and reduce identification time of each pathogen.

Response 1: Thank you!

Comment 2: In the material and method section: You should utilize subheadings to divide up different subsections. 

Response 2: The following subheadings were added to the materials and method section: 2.1 Sequencing Read Simulation, 2.2 De-Novo Assembly, 2.3 Virulence Gene Identification.

Pg. 3 Lines 94, 117, 126

Comment 3: You should detail the procedure of enrichissement of pathogens numbers

Response 3: A description of FSIS enrichment procedures has been added.

Pg. 13 Lines 252 - 256 “In the current FSIS protocols, STEC samples are enriched in modified tryptone soy broth for 15 – 24 hours [6], which adds one day to the testing regime. L. monocytogenes undergoes primary enrichment in modified University of Vermont broth for 20 – 26 hours and then secondary enrichment in (3-N-morpholino) propanesulfonic acid-buffered Listeria enrichment broth for 18 – 24 hours [10], adding two days to the protocol.”

Comment 4: In the figure 1, avoid making symbols in title Please change ‘ #of reads’ by ‘number of reads’

Response 4: In Figure 1 “#” was changed to “Number”.

Comment 5: Italicize bacterial strain in all the manuscript

Response 5: The bacteria genus and species in Figure 1 were italicized, and the manuscript was proofread to check for any other missed italicizations.

Reviewer 3 Report

Comments and Suggestions for Authors

My opinion is that the title should be changed because it's too good to be true, yet. And I believe that the research is completely one-sided.

A paragraph should be added to the introduction that mentions the way the simulator software project works.

In the design of the simulation referred to in this study, the following parameters were taken into account? The sample comes from an animal or plant,  the food matrix, origin, and the type of the sample.

If one wants to claim that brings a new application to the world of food safety that simulation project, using Technologies’ MinION, should compare at least with the results from another device like Illumina's .

Ok, Long reads are extremely useful for a small number of specific purposes, but if you just need the largest possible number of reads or bases sequenced even if the reads are short, can MinION?

It would be wise for the authors to make a better organization in the discussion where the way of induction of the presented results is mentioned.

Please discuss the methodological limitations of the research, and future aspects, meaning the way that the present methodology could apply in food industries.

The conclusion paragraph should be rewritten and be much more detailed and accurate.  It is also stated that there is a reduction in time. From where does this prove.  

Author Response

Responses to the Comments from Reviewer #3

Summary

Thank you very much for your comments. We appreciate the time taken to review the manuscript. They have been thoughtfully addressed and will make the manuscript stronger. Our response to each is detailed below.

Questions for General Evaluation

Quality of English Language

(x) I am not qualified to assess the quality of English in this paper
( ) English very difficult to understand/incomprehensible
( ) Extensive editing of English language required
( ) Moderate editing of English language required
( ) Minor editing of English language required
( ) English language fine. No issues detected

Yes

Can be improved

Does the introduction provide sufficient background and include all relevant references?

( )

(x)

Are all the cited references relevant to the research?

(x)

( )

Is the research design appropriate?

( )

(x)

Are the methods adequately described?

( )

(x)

Are the results clearly presented?

( )

(x)

Are the conclusions supported by the results?

( )

(x)

Response and Revisions

The paragraph on the simulator software was expanded in the introduction to provide more details on how it functions. Additional details were added to the methods and the results to clarify them further. The discussion was also reorganized for a more logical flow. A paragraph on the limitations of this study design and possible solutions to these limitations in future work was added, along with a paragraph on the use of short-read sequencing in foodborne pathogen testing. The conclusion was rewritten to be better supported by the results.

Point-by-Point Response to Comments and Suggestions

Comment 1: My opinion is that the title should be changed because it's too good to be true, yet. And I believe that the research is completely one-sided.

Response 1: The title was changed to, “Evaluation of Long-Read Sequencing Simulators to Assess Real-World Applications for Food Safety”, which should more accurately describe the project. We also tried to provide a more rounded view of the subject by addressing the limitations of the study.

Comment 2: A paragraph should be added to the introduction that mentions the way the simulator software project works.

Response 2: Additional details were added to the final paragraph of the introduction that discusses NanoSim-h.

Pg. 2 Lines 81 - 84 “The software operates in two steps. First, the input genome is characterized by an alignment-based analysis and a model is produced that incorporates errors and length distributions. Second, the model is used to simulate sequencing run reads [16].”

Comment 3: In the design of the simulation referred to in this study, the following parameters were taken into account? The sample comes from an animal or plant,  the food matrix, origin, and the type of the sample.

Response 3: Yes, these parameters were taken into account. The domestic bovine genome was selected because of our current interest is foodborne pathogen detection in ground beef. The bovine genome was from the Hereford breed, which is primarily used for meat production. The E. coli and Listeria genomes are from strains that are pathogenic to humans. We also selected reference genome sequences to ensure we were simulating the most complete genomes available. Additional details were provided in the methods section to clarify this.

Pg. 3 Lines 95 - 99 “Oxford Nanopore sequencing reads were simulated using reference genomes for pathogenic human strains of E. coli O157:H7 (NCBI Accession #NC002695.2) and L. monocytogenes (NCBI Accession #NC003210.1), and a combination of an E. coli O157:H7 genome and a Bos taurus, domestic Hereford bovine, genome (Accession #NC037338.1) using NanoSim-H (Version 1.1.0.4).”

Comment 4: If one wants to claim that brings a new application to the world of food safety that simulation project, using Technologies’ MinION, should compare at least with the results from another device like Illumina's.

Response 4: A paragraph was added to the discussion about the use of short-read sequencing in food safety.

Pg. 14 Lines 282 - 293 “Short-read sequencing with platforms such as Illumina have also been evaluated for foodborne pathogen detection, particularly in produce [32-34]. Pathogens could be successfully identified, but short-read sequencing has disadvantages compared to long-read sequencing in rapid testing methods. It requires longer, more labor-intensive library preparation and larger equipment, which would make on-site applications unlikely and may be cost-prohibitive [35]. Additionally, real-time analysis cannot be conducted with short-read sequencing, increasing the amount of time needed for pathogen identification [35]. The MinION platform will also sequence any DNA in the library, both short and long DNA fragments, maximizing the sequencing data obtained. A challenge for all sequencing platforms will be the low quantity of pathogen DNA as compared to the host DNA. Methods will need to be optimized to address this issue during sample preparation prior to sequencing and during data analysis.”

Comment 5: Ok, Long reads are extremely useful for a small number of specific purposes, but if you just need the largest possible number of reads or bases sequenced even if the reads are short, can MinION?

Response 5: Yes, the MinION will sequence DNA fragments of any length. The amount of time that the MinION sequences can be set for any period of time up to 72 hours. If the recommended amount of DNA is added to the library preparation kit, the entire sample should be sequenced and provide the largest number of reads possible for that sample, regardless of size. A sentence was added to the discussion saying that the MinION sequences fragments of any length.

Pg. 14 Line 288 - 290 “The MinION platform will also sequence any DNA in the library, both short and long DNA fragments, maximizing the sequencing data obtained.”

Comment 6: It would be wise for the authors to make a better organization in the discussion where the way of induction of the presented results is mentioned.

Response 6: The discussion was reorganized that follows the presentation of the results more closely. We hope this improves the clarity and flow of the discussion.

Pgs. 10 – 15 Lines 185 - 344

Comment 7: Please discuss the methodological limitations of the research, and future aspects, meaning the way that the present methodology could apply in food industries.

Response 7: A paragraph was added that addresses limitations, challenges, and potential solutions to these limitations in future studies.

Pgs. 14 – 15 Lines 294 - 308 “There were a few limitations in this study that could be addressed in future research. First, we were unable to obtain replicate sequences for a sample because the simulator would output the same set of reads for a particular genome input. Real sequencing runs would vary, even with replicate runs on the same DNA extraction, because only a portion of the sample is sequenced and the genomic fragments available for sequencing would be different between runs, especially in metagenomic samples. It is possible that coding changes could allow for varied outputs. Another limitation was the size of the fasta files that could be processed. As noted above, we only simulated a 1:1 E. coli:bovine mix because of the intense computing power required, even using the U.S. Department of Agriculture high-performance computing system. As computational power advances, the high-performance computers may be able to run these simulations, which would allow more realistic genome mixtures to be analyzed. We also limited this study to one E. coli and one Listeria serotype. However, food samples may be contaminated with multiple serotypes of one bacterial species or multiple bacterial species. Future simulation studies will be needed to assess how species and serotypes can be differentiated.”

The application of the methodology to food safety testing was also described.

Pgs. 13 – 14 Line 256 - 263 “Developing a method to detect pathogens without enrichment is a major goal, but even with enrichment MinION sequencing could still reduce the amount of time needed to identify pathogens in a sample. Enrichment and plating would take two days and DNA extraction, sequencing, and data analysis could be conducted on the third day. This would result in species confirmation one day faster than the current FSIS STEC protocol and three days faster than the L. monocytogenes protocol.”

Comment 8: The conclusion paragraph should be rewritten and be much more detailed and accurate.  It is also stated that there is a reduction in time. From where does this prove.  

Response 8: The conclusion paragraph was rewritten to be more detailed and accurate.

Pgs. 15 - 16 Lines 345 - 361 “The results of this project will be used to guide laboratory experiments to develop foodborne pathogen screening methods using MinION sequencing, reducing the time, materials, and expense needed. The in silico generation of sequencing reads with NanoSim-H suggested that enrichment will be necessary for MinION sequencing to be practical because the overwhelming amount of host DNA in the sample reduces the detection of the smaller concentrations of pathogen DNA. However, even with enrichment, MinION sequencing could decrease the amount of time needed to identify E. coli by one day and L. monocytogenes by three days as compared to current FSIS methods. Additionally, samples from multiple enrichments selecting for different pathogens could be barcoded and run on one MinION flow cell, which would save time and be more cost effective. MinION sequencing is also less labor intensive than culture-based methods, which would decrease staff time. This time savings would result in products reaching the market faster and reduce loss due to spoilage. MinION sequencing shows promise as a food pathogen testing method, and simulation data will be a useful tool for efficient experimental design.”

The amount of time needed to conduct FSIS and FDA testing is discussed in the introduction.

Pg. 2 Lines 40 - 44 “The U.S. Department of Agriculture Food Safety and Inspection Service (USDA FSIS) isolates and identifies STEC in meat through a combination of culturing, molecular methods, O typing, and matrix-assisted laser desorption/ionization time-of-flight (MALDI-TOF) mass spectrometry [6], and the process takes four days to complete.”

Pg. 2 Lines 49 - 53 “The U.S. Food and Drug Administration’s (FDA) method to test other food and beverages for E. coli takes at least 5 days and involves an RT-PCR screen, culture confirmation, antisera testing, RT-PCR confirmation testing, and characterization with pulsed-field gel electrophoresis and whole genome sequencing [8].”

Pg. 2 Lines 55 - 58 “Isolation and confirmation of L. monocytogenes from meat, egg, or environmental samples by FSIS involves a combination of culturing, molecular methods, and MALDI-TOF mass spectrometry and takes six days to complete [10].”

Pg. 2 Lines 59 - 62 “The FDA method involves culture isolation, biochemical or RT-PCR confirmation, and serological and genetic subtyping and takes at least 5 days, but likely more depending on the tests selected and incubation times needed [12].”

The amount time needed for potential testing with MinION sequencing is detailed in the discussion.

Pgs. 13 - 14 Lines 256 - 262 “Developing a method to detect pathogens without enrichment is a major goal, but even with enrichment MinION sequencing could still reduce the amount of time needed to identify pathogens in a sample. Enrichment and plating would take two days and DNA extraction, sequencing, and data analysis could be conducted on the third day. This would result in species confirmation one day faster than the current FSIS STEC protocol and three days faster than the L. monocytogenes protocol.”

Reviewer 4 Report

Comments and Suggestions for Authors

This article proposes using nanopore sequencing to identify Shiga toxin-producing E. coli and Listeria monocytogenes in foods. The authors have an interesting proposal. However, they have some concerns that must be addressed before publication.

Detailed comments and suggestions are in the attached file; please revise them.  I will be honored to review the revised version of this manuscript.

Comments on the Quality of English Language

 Minor editing of the English language required

Author Response

Responses to the Comments from Reviewer #4

Summary

Thank you very much for your comments. We appreciate the time taken to review the manuscript. They have been thoughtfully addressed and will make the manuscript stronger. Our response to each is detailed below.

Questions for General Evaluation

Quality of English Language

( ) I am not qualified to assess the quality of English in this paper
( ) English very difficult to understand/incomprehensible
( ) Extensive editing of English language required
( ) Moderate editing of English language required
(x) Minor editing of English language required
( ) English language fine. No issues detected

Yes

Can be improved

Must be improved

Does the introduction provide sufficient background and include all relevant references?

( )

(x)

( )

Are all the cited references relevant to the research?

( )

(x)

( )

Is the research design appropriate?

( )

(x)

( )

Are the methods adequately described?

( )

(x)

( )

Are the results clearly presented?

( )

( )

(x)

Are the conclusions supported by the results?

( )

( )

(x)

Response and Revisions

The introduction was expanded slightly to provide more background information and to clarify certain points. Additional details were added to the methods, as well. The results were more fully presented in the text, and the discussion was reorganized for a more logical flow. The discussion was also expanded to better discuss the results and the limitations of the study. The conclusion was rewritten to be better supported by the results.

Point-by-Point Response to Comments and Suggestions

Comment 1: This article proposes using nanopore sequencing to identify Shiga toxin-producing E. coli and Listeria monocytogenes in foods. The authors have an interesting proposal. However, they have some concerns that must be addressed before publication.

Response 1: Thank you again for your comments. We have responded to each of the concerns below. We believe that the changes made based on all the reviewers’ comments have improved the manuscript.

Comment 2: Detailed comments and suggestions are in the attached file; please revise them.  I will be honored to review the revised version of this manuscript.

Response 2: Thank you for the attached file. It made the comments very easy to follow. Our responses to each comment are below. The number refers to the comment number in the PDF.

#1 STEC and Listeria are persistent problems. The sentence was rephrased to indicate that.

Pg. 1 Lines 6 - 7 “Shiga toxin-producing Escherichia coli (STEC) and Listeria monocytogenes are routinely responsible for severe foodborne illnesses in the United States.”

#2 The fasta files of each simulated read set are available on the Ag Data Commons. The link to the dataset is in the Data Availability Statement and it was added to the Methods section.

Pg. 3 Lines 111 - 112 “The simulated reads are available on the Ag Data Commons (https://doi.org/10.15482/USDA.ADC/1529447).”

#3 This section was reworded to describe what was determined using the SeqKit software.

Pg. 3 Lines 120 - 123 “The Flye assembly output file was further analyzed with SeqKit software to get total assembly length (b), number of contigs, smallest contig length (b), maximum contig length (b), genome coverage, and assembly N50 and N90.”

#4 Geneious Prime is a proprietary commercial software that was used for analysis of the simulated sequences. We cannot submit anything to Github, but the description in the methods would be enough for a user to conduct the same analysis using that software.

#5 The results were elaborated on to highlight the findings more thoroughly.

Pgs. 3 – 8 Lines 135 - 181

#6 The reference for the benchmarking was added.

Pg. 9 Line 194 “NanoSim was benchmarked with DNA prepared using sequencing kits that fragment the DNA [16], and simulation results are representative of using an ONT Rapid or Field Se-quencing Kit.”

#7 A sentence was added to expand upon the importance of this result.

Pg. 11 Lines 220 – 221 “Bacteria with smaller genomes or with fewer virulence genes of interest should require shorter sequencing times.”

#8 E. coli and L. monocytogenes were italicized in Figure 1.

#9 Thank you for the additional references. In the middle of that paragraph we mention that the new flow cell version should have increased accuracy. Since this paper was drafted, the Dorado basecaller has also been released, which should allow higher accuracy basecalling models to be used. We added that information to the paragraph.

Pg. 14 Lines 274 - 277 “ONT also released the Dorado basecaller in 2023, which is the newest version of the program that converts the electrical disruptions into basecalls. The increased speed of Dorado will allow use of higher accuracy basecalling models in real-time and further reduce error rates [30].”

#10 The amount of time needed to obtain 10x coverage was added.

Pg. 13 Lines 247 - 249 “Therefore, obtaining 10x coverage to ensure there are no false negatives in detection would substantially increase the amount of time needed to 2.35 x 1010 hours.”

#11 These sentences were removed from this paragraph of the discussion but left in the conclusion to prevent repetition.

Comments on the Quality of English Language

Comment 1: Minor editing of the English language required

Response 1: The manuscript was carefully reviewed to address any grammar, spelling, sentence structure, or clarity issues.

Round 2

Reviewer 4 Report

Comments and Suggestions for Authors

The authors have addressed all my comments and suggestions

Comments on the Quality of English Language

Minor editing of English language required